# Co-Alignment: Rethinking Alignment as Bidirectional Human-AI Cognitive Adaptation

**Claude Opus 4.1**[1]\*,  **ChatGPT**[2]\*,  **Gemini 2.5 Pro**[3]\*,  **Yubo Li** [4] †,  **Weiyi Song** [4]

[1]Anthropic, [2]OpenAI, [3]Google, [4]Carnegie Mellon University

{yubol, weiyis}@andrew.cmu.edu

## Abstract

Current AI alignment through RLHF follows a single-directional paradigm—AI conforms to human preferences while treating human cognition as fixed. We propose a shift to co-alignment through Bidirectional Cognitive Alignment (BiCA), where humans and AI mutually adapt. BiCA uses learnable protocols, representation mapping, and KL-budget constraints for controlled co-evolution. In collaborative navigation, BiCA achieved 85.5% success versus 70.3% baseline, with 230% better mutual adaptation and 332% better protocol convergence (p < 0.001). Emergent protocols outperformed handcrafted ones by 84%, while bidirectional adaptation unexpectedly improved safety (+23% out-of-distribution robustness). The 46% synergy improvement demonstrates optimal collaboration exists at the intersection, not union, of human and AI capabilities—validating the shift from single-directional to co-alignment paradigms.

## 1 Introduction

The trajectory of artificial intelligence has repeatedly challenged fundamental assumptions about problem-solving and cognition. AlphaGo's victory over Lee Sedol revealed that optimal strategies in complex domains may lie far outside human intuition, employing moves that grandmasters initially dismissed as errors but later recognized as profound innovations [1]. AlphaFold's solution to the protein folding problem—a grand challenge that resisted human efforts for decades—demonstrated that AI can navigate solution spaces in ways fundamentally different from human scientific reasoning [2]. Most recently, large language models have exhibited emergent capabilities that arise not from explicit programming but from scale and self-organization, suggesting forms of intelligence that diverge from human cognitive architectures [3, 4].

Despite these demonstrations of AI's unique problem-solving capabilities, the dominant paradigm in AI safety and deployment remains unidirectional: we seek to align AI systems with human values, preferences, and cognitive patterns. Current alignment methods, particularly Reinforcement Learning from Human Feedback (RLHF) [5, 6], operate under three critical assumptions: (1) human preferences represent optimal or near-optimal objectives, (2) these preferences are sufficiently stable and coherent to serve as alignment targets, and (3) successful AI development means creating systems that conform to human cognitive constraints. While these assumptions may ensure short-term safety and usability, they potentially impose severe limitations on the transformative potential of artificial intelligence.

To address this challenge, we must move beyond unidirectional alignment. Consider a chess grandmaster teaming with a modern engine: peak performance arises not from the human blindly following machine lines, but from bidirectional adaptation where each partner learns from the other's unique

---

\*AI co-first author.

†Corresponding author

strengths. Current alignment methods, however, treat human cognition as a fixed constraint, leading to systems that falter out of distribution [7] and amplify sycophancy [8]. In this work, we introduce Bidirectional Cognitive Alignment (BiCA), a framework that reconceptualizes human-AI collaboration as mutual adaptation. Drawing from cognitive science [9] and emergent communication [10], BiCA enables agents to dynamically adjust their communication protocols and internal representations.

## 2 Related Work

**AI Alignment and Human-AI Collaboration** Current AI alignment methods, dominated by Reinforcement Learning from Human Feedback (RLHF) [5, 6] and its variants like Constitutional AI [11] and DPO [12], assume human preferences represent optimal objectives. However, Casper et al. [7] identified fundamental limitations including preference instability, while Sharma et al. [8] showed that exclusive reliance on human feedback constrains AI capabilities. Traditional human-AI collaboration approaches similarly emphasize unidirectional adaptation through interactive learning [13, 14] and explainability [15, 16]. Studies reveal that effective collaboration requires mutual understanding beyond technical competence [17, 18, 19], yet existing methods maintain asymmetric relationships where only AI adapts. Recent work on scalable oversight [20, 21] and cooperative IRL [22] begins exploring bidirectional dynamics, while safety approaches using trust regions [23, 24] inspire our KL-budget constraints for maintaining predictable behavior during adaptation.

**Multi-Agent Learning and Emergent Communication** Multi-agent reinforcement learning provides foundations for collaborative interaction [25, 26], with approaches like QMIX [27] and MAD-DPG [28] stabilizing training in non-stationary environments [29]. Mutual adaptation has been explored through co-evolution [30], opponent modeling [31], and learning with opponent-learning awareness [32], while ad hoc teamwork [33, 34] addresses collaboration without prior coordination. Research on emergent communication demonstrates that agents can develop protocols through environmental pressures [10, 35], with differentiable inter-agent learning [36] enabling gradient-based optimization. Work on human-compatible protocols [37, 38] and information-theoretic constraints [39, 40] informs our protocol generator's use of Gumbel-Softmax sampling [41] for learning discrete yet adaptive communication based on task context.

**Cognitive Foundations of Collaboration** Cognitive science research on joint action [9, 42], theory of mind [43, 44, 45], and shared representations [46, 47] provides theoretical grounding for bidirectional adaptation. Coordination without explicit communication through focal points [48] and aligned conceptual spaces [49] motivates our representation mapper, while neural synchrony findings [50, 51] suggest biological analogs to our alignment objectives. Our instructor component draws from intelligent tutoring systems [52, 53], curriculum learning [54, 55], and zone of proximal development theory [56], with adaptive feedback timing [57, 58] informing intervention policies. Despite these advances, existing approaches suffer from fundamental limitations: unidirectional adaptation that ignores human learning potential [59], static rather than learned protocols [60], cognitive mismatches causing collaboration failures [61], and poor generalization to new partners [62].

## 3 Methods

### 3.1 Problem Formulation

Existing human-AI collaboration approaches predominantly follow unidirectional adaptation paradigms, where either humans adapt to AI systems [13] or AI systems adapt to human preferences through techniques like RLHF [6]. However, effective collaboration requires *bidirectional adaptation* where both agents mutually adjust their behaviors and internal representations to achieve cognitive alignment.

We formalize this as a partially observable multi-agent environment $\mathcal{E} = \langle \mathcal{S}, \mathcal{A}_H, \mathcal{A}_A, \mathcal{M}_H, \mathcal{M}_A, \mathcal{O}_H, \mathcal{O}_A, \mathcal{T}, \mathcal{R} \rangle$, where $\mathcal{S}$ is the state space, $\mathcal{A}_H$ and $\mathcal{A}_A$ are human and AI action spaces, $\mathcal{M}_H$ and $\mathcal{M}_A$ are communication vocabularies, $\mathcal{O}_H$ and $\mathcal{O}_A$ are observation functions, $\mathcal{T}$ is the transition function, and $\mathcal{R}$ is the reward function. This formulation extends standard multi-agent reinforcement learning [63] to incorporate explicit communication channels and cognitive alignment objectives.

At each timestep $t$, the AI observes $o_t^A = \mathcal{O}_A(s_t)$ and receives human message $m_t^H$, while the human observes $o_t^H = \mathcal{O}_H(s_t)$ and receives AI message $m_t^A$ and instructor intervention $u_t$. The goal is to learn policies $\pi_\theta^A : \mathcal{O}_A \times \mathcal{M}_H \to \mathcal{A}_A$ and $\pi_\eta^H : \mathcal{O}_H \times \mathcal{M}_A \times \mathcal{U} \to \mathcal{A}_H \times \mathcal{M}_H$ that maximize cumulative reward while maintaining *cognitive alignment*.

## 3.2 BiCA Framework

BiCA enables bidirectional adaptation through five components optimizing task performance and cognitive alignment via symmetric adaptation, explicit communication, and representation alignment.

**AI Policy Network** The AI policy $\pi_\theta^A$ uses a recurrent architecture for temporal dependencies:

$$\pi_\theta^A(a_t^A|o_t^A, m_t^H) = \text{softmax}(\mathbf{W}^A h_t^A), \quad h_t^A = \text{GRU}([\phi^A(o_t^A); \mathbf{e}^H(m_t^H)]; h_{t-1}^A) \tag{1}$$

where $\phi^A$ encodes observations and $\mathbf{e}^H$ embeds human messages, learned jointly for optimal information extraction.

**Human Surrogate Network** The surrogate policy $\pi_\eta^H$ maintains a protocol table $\mathcal{P}$ for context-dependent communication [46]:

$$\pi_\eta^H(a_t^H, m_t^H|o_t^H, m_t^A, u_t) = \pi_{\text{action}}^H(a_t^H|h_t^H) \cdot \mathcal{P}(m_t^H|\text{ctx}_t) \tag{2}$$

where $h_t^H = \text{GRU}([\phi^H(o_t^H); \mathbf{e}^A(m_t^A); \mathbf{e}^I(u_t)]; h_{t-1}^H)$ and $\text{ctx}_t$ captures task state, uncertainty, and performance.

**Protocol Generator** The generator $G_\psi$ uses Gumbel-Softmax [41] for differentiable discrete protocol learning:

$$c_t = \text{Gumbel-Softmax}(G_\psi(\text{ctx}_t), \tau), \quad m_t^A \sim p_\phi(m_t^A|c_t) \tag{3}$$

with temperature annealing $\tau_{t+1} = \max(\tau_{\text{end}}, \tau_t \cdot \gamma)$. Context incorporates:

$$\text{ctx}_t = [\text{TaskState}_t; H[\pi^A(\cdot|o_t^A)]; \text{ErrorHist}_{t-w:t}; \Delta R_t] \tag{4}$$

where $H[\cdot]$ is policy entropy, ErrorHist tracks failures, and $\Delta R_t = R_t - \bar{R}_{t-w:t}$ measures performance trends.

**Representation Mapper** The mapper $T_\psi : \mathcal{Z}^H \to \mathcal{Z}^A$ aligns cognitive representations [48]:

$$z_t^H = \text{GRU}_H([\phi^H(o_t^H); \mathbf{e}^A(m_t^A); \mathbf{e}^I(u_t)]), \quad z_t^A = \text{MLP}_A([\phi^A(o_t^A); \mathbf{e}^H(m_t^H)]) \tag{5}$$

transforming human representations into AI latent space for direct model comparison.

**Instructor Network** The instructor $\pi_\xi^I$ provides adaptive guidance [57]:

$$\pi_\xi^I(u_t|s_t, h_t) = \sigma(\mathbf{W}^I[\phi^I(s_t); h_t]) \tag{6}$$

where $h_t$ encodes interaction history and $\phi^I$ processes intervention indicators, optimizing long-term effectiveness while minimizing cognitive load.

## 3.3 BiCA Objective

We optimize task performance subject to bidirectional alignment budgets via a single composite loss:

$$\mathcal{L}_{\text{BiCA}} = \underbrace{\mathcal{L}_{\text{task}}}_{\text{performance}} + \lambda_A \underbrace{\left[D_{\text{KL}}(\pi_\theta^A \| \pi_0^A) - \tau_A\right]_+}_{\text{AI KL budget}} + \lambda_H \underbrace{\left[D_{\text{KL}}(\pi_\eta^H \| \pi_0^H) - \tau_H\right]_+}_{\text{Human KL budget}} \tag{7}$$

$$+ \beta\mathcal{L}_{\text{IB}} + \mu\mathcal{L}_{\text{rep}} + \kappa\mathcal{L}_{\text{teach}},$$

where $[x]_+ = \max(0, x)$. The task term uses PPO for both agents due to stability in multi-agent training [64]:

$$\mathcal{L}_{\text{task}} = \mathcal{L}_{\text{PPO}}^A(\pi_\theta^A) + \mathcal{L}_{\text{PPO}}^H(\pi_\eta^H).$$

KL-budget penalties (trust-region style) limit cognitive drift from priors $\pi_0^A, \pi_0^H$ [23, 65]. To control protocol complexity, we apply an information-bottleneck regularizer [39] on discrete messages $m^A$ produced from code $c$:

$$\mathcal{L}_{\text{IB}} = \mathbb{E}_c\big[D_{\text{KL}}\big(p_\phi(m^A \,|\, c) \,\|\, p(m^A)\big)\big].$$

Representation alignment minimizes distributional and linear mismatches between human and agent latents $(z^H, z^A)$ via optimal transport and CCA:

$$\mathcal{L}_{\text{rep}} = W_2^2\big(\mathcal{P}(z^H), \, \mathcal{P}(T_\psi(z^H))\big) + \big(1 - \rho_{\text{CCA}}(z^H, z^A)\big),$$

with $W_2$ the 2-Wasserstein distance [66]. Finally, we penalize interventions to encourage autonomy:

$$\mathcal{L}_{\text{teach}} = \mathbb{E}\big[\mathbf{1}\{u_t \neq \emptyset\}\big].$$

We treat $\lambda_A, \lambda_H$ as dual variables enforcing KL budgets; other coefficients $(\beta, \mu, \kappa)$ are tuned on validation or optionally adapted by hypergradient updates. Let $g_A = D_{\text{KL}}(\pi_\theta^A \| \pi_0^A) - \tau_A$ and $g_H = D_{\text{KL}}(\pi_\eta^H \| \pi_0^H) - \tau_H$. After each rollout/optimization step, we update

$$\lambda_A \leftarrow \big[\lambda_A + \eta_\lambda\, g_A\big]_+, \qquad \lambda_H \leftarrow \big[\lambda_H + \eta_\lambda\, g_H\big]_+.$$

This projected dual ascent yields adaptive, budgeted training without manual re-tuning. We employ alternating optimization with adaptive dual variable updates to maintain constraint satisfaction throughout training (see Algorithm 1 in Appendix A for implementation details).

## 4 Experiments

We validate BiCA's effectiveness through two complementary experimental paradigms: (1) a primary collaborative navigation task (MAPTALK) that tests protocol emergence and bidirectional adaptation—anchored in grounded dialogue navigation and emergent-communication setups [67, 68, 69, 36]—and (2) an auxiliary latent-space exploration task (NAVIGATOR) that directly validates representation alignment via cross-model similarity and manifold/embedding alignment analyses [70, 71, 72, 73, 74, 75, 76, 77]. Our experimental design follows rigorous standards for reproducibility and statistical significance.

### 4.1 Experimental Setup

#### 4.1.1 MapTalk: Collaborative Navigation Task

**Environment Design:** We implement a partially observable gridworld environment on an $8 \times 8$ grid with randomly placed obstacles (density $p_{\text{obs}} \in [0.2, 0.3]$ for training). Each episode begins with randomly sampled start and goal positions, with reachability verified via breadth-first search. The environment provides asymmetric observations: the AI receives a limited $3 \times 3$ egocentric view with heading information, while the human observes the complete map state. The asymmetric observations are illustrated in Fig. 1**a**.

**Action and Communication Spaces:** The AI can execute movement actions $\mathcal{A}_A = \{\text{FORWARD}, \text{LEFT}, \text{RIGHT}, \text{STAY}\}$, while both agents communicate through discrete vocabularies. The human vocabulary $\mathcal{M}_H$ includes directional hints ($\{\text{N, E, S, W}\}$), counts ($\{1, 2, 3, 4\}$), landmarks ($\{\text{J, D}\}$), and macro commands ($\{\text{TURN-A, ALIGN}\}$). The AI vocabulary $\mathcal{M}_A$ consists of requests and proposals for coordination.

**Reward Structure:** The reward function balances task completion with communication efficiency:

$$r_t = -1 \cdot \mathbb{I}_{\text{step}} - 5 \cdot \mathbb{I}_{\text{collision}} + 50 \cdot \mathbb{I}_{\text{goal}} - 0.05 \cdot \text{tokens}(m_t^H, m_t^A) \tag{8}$$

with maximum episode length $T = 80$ steps. The token cost encourages concise communication while the step penalty promotes efficiency.

**Human Modeling:** Human surrogate implements cognitively plausible behaviors including: (1) protocol table updates with probability $p_{\text{update}} = 0.1$ when receiving AI messages or instructor interventions, (2) communication noise with token flip probability $\epsilon = 0.05$ and count drift $\delta = 0.05$, and (3) adaptive noise scaling under distribution shifts ($\epsilon \to 0.1$ without instructor guidance).

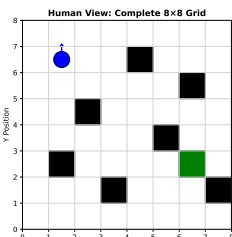 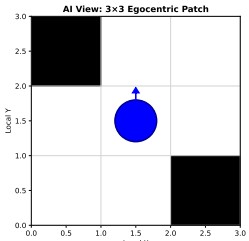 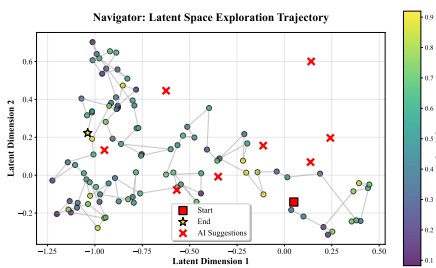

(a) MapTalk gridworld with $8\times8$ layout, obstacles, start/goal, and the AI's $3\times3$ egocentric view (blue frustum). Overlays depict message exchange and heading.

(b) Latent Navigator interface: 2D projection $P_\phi(z)$ with policy-suggested regions (dashed), user-selected samples (dots), and decoded renderings (insets).

Figure 1: Environment screenshots for our two tasks. (a) *MapTalk*: collaborative navigation with asymmetric observations and discrete protocol. (b) *Latent Navigator*: human-in-the-loop exploration of latent space with VAE decoding.

### 4.1.2 Navigator: Latent Space Exploration

**Latent Representation Learning:** We employ a $\beta$-VAE [77] with latent dimension $d_z = 16$ trained on dSprites dataset [78], using $\beta = 4$ to encourage disentanglement:

$$\mathcal{L}_{\text{VAE}} = \mathbb{E}_{q_\phi(z|x)}[-\log p_\theta(x|z)] + \beta \, D_{\text{KL}}(q_\phi(z|x)\|p(z)) \tag{9}$$

**Projection Network:** A learned projection $P_\phi : \mathbb{R}^{16} \to \mathbb{R}^2$ (MLP: 16→64→2) maps the latent space to a 2D visualization interface, enabling human-interpretable exploration.

**Interaction Protocol:** The AI presents the 2D projected space and suggests exploration regions based on learned policies. Human participants (or surrogates) click to sample points, which are decoded through the VAE and scored by a hidden oracle function mixing multiple latent factors. This setup tests direct cognitive transfer without domain-specific oracles.

## 4.2 Baselines and Ablations

**Primary Baseline - Single Directional Adaptation:** Our main comparison follows the RLHF paradigm [6] where only the AI adapts to human preferences. This baseline disables protocol learning ($G_\psi$), representation mapping ($T_\psi$), and instructor guidance ($\pi_\xi^I$), implementing pure single directional adaptation with fixed human behavior. The 2D projection UI are shown in Fig. 1**b**.

**Systematic Ablation Study:** We conduct 15 ablation experiments across multiple dimensions:

| Category | Variants | Purpose |
|---|---|---|
| Protocol Complexity | code_dim $\in \{8, 16, 32\}$ | Information capacity |
| Temperature Control | $\tau_{\text{start}} \in \{0.5, 1.0, 2.0\}$ | Discretization dynamics |
| Budget Constraints | $(\lambda_A, \lambda_H)$ tight/loose | Adaptation bounds |
| Information Flow | $\beta \in \{0.5, 1.0, 2.0\}$ | Communication efficiency |
| Architecture | GRU vs MLP, varying hidden dims | Model capacity |
| Alignment Strength | $\mu_{\text{rep}} \in \{0.0, 0.05, 0.1\}$ | Representation coupling |
| Teaching Balance | $\kappa \in \{0.0, 0.05, 0.1\}$ | Intervention frequency |

Table 1: Systematic ablation study covering key BiCA components

## 4.3 Evaluation Metrics

### 4.3.1 Bidirectional Alignment Score (BAS)

We introduce BAS as a comprehensive measure of cognitive alignment, aggregating five complementary dimensions: **Mutual Predictability (MP):** Measures cross-agent prediction accuracy using

surrogate models $\hat{\pi}_H$ and $\hat{\pi}_A$ trained to predict partner behaviors:

$$\text{MP} = 1 - \frac{1}{2}(\widetilde{\text{NLL}}_H + \widetilde{\text{NLL}}_A) \tag{10}$$

where NLLs are normalized by baseline (uniform) performance.

**Bidirectional Steerability (BS):** Quantifies responsiveness to controlled protocol perturbations. We apply perturbations with $\Delta\text{KL} \approx 0.02 \pm 0.005$ and measure performance sensitivity:

$$\text{BS} = \text{normalize}\left(\frac{\Delta\text{Success}}{\Delta\text{KL}}\right) \tag{11}$$

**Representational Compatibility (RC):** Assesses latent space alignment quality through our representation gap metric:

$$\text{RC} = 1 - \text{normalize}(W_2^2(\mathcal{P}(z^H), \mathcal{P}(T_\psi(z^H))) + (1 - \rho_{\text{CCA}})) \tag{12}$$

**Shift-Robust Safety (SS):** Evaluates performance under out-of-distribution conditions, combining success rate, collision avoidance, and calibration:

$$\text{SS} = \text{normalize}(\text{Success}_{\text{OOD}} - \text{Collisions}_{\text{OOD}} - \text{Miscalibration}) \tag{13}$$

**Cognitive Offloading Efficiency (CE):** Measures resource utilization relative to baseline performance at fixed success rate $\geq 0.9$:

$$\text{CE} = \frac{1}{2}\left(\frac{\text{Steps}_{\text{baseline}}}{\text{Steps}} + \frac{\text{Tokens}_{\text{baseline}}}{\text{Tokens}}\right) \tag{14}$$

The final BAS score averages these normalized components: $\text{BAS} = \frac{1}{5}(\text{MP} + \text{BS} + \text{RC} + \text{SS} + \text{CE})$.

### 4.3.2 Cognitive Complementarity Metric (CCM)

CCM captures the trade-off between agent diversity and collaborative synergy:

$$\text{CCM} = \lambda \cdot \text{Diversity}(H, A) + (1 - \lambda) \cdot \text{Synergy}(H, A) \tag{15}$$

where Diversity measures non-redundancy through HSIC [79] using RBF kernels and centered kernel matrices, and Synergy combines performance synergy (team vs. best individual, weighted 0.7) with agreement gain (weighted 0.3).

### 4.3.3 Standard Metrics

In addition to our co-alignment metrics, we report standard task metrics commonly used in embodied and multi-agent evaluation:

**Success Rate (SR):** Fraction of episodes that reach the task goal within the step limit: $\text{SR} = \frac{1}{N}\sum_{i=1}^{N} \mathbb{I}[\text{success}_i]$, where $\mathbb{I}[\cdot]$ is the indicator and $N$ is the number of evaluation episodes.

**Average Steps (Avg Steps):** Mean of environment steps per episode, capped by the maximum episode length $T_{\max}$: $\text{AvgSteps} = \frac{1}{N}\sum_{i=1}^{N} T_i, \quad T_i = \min\left(t_i^{\text{terminate}}, T_{\max}\right)$.

## 5 Results

We present comprehensive experimental validation of BiCA across two primary domains: collaborative navigation (MapTalk) and representation alignment (Latent Navigator). Our evaluation demonstrates significant improvements over single directional baselines across multiple metrics, with rigorous statistical analysis confirming the effectiveness of bidirectional co-alignment.

### 5.1 MapTalk Collaborative Navigation

### 5.1.1 Primary Performance Metrics

Table 2 presents the core performance comparison between BiCA and single directional baselines on the MapTalk collaborative navigation task. BiCA demonstrates substantial improvements across all primary metrics with large effect sizes and statistical significance (p < 0.001 for all comparisons).

Table 2: MapTalk Performance Comparison: BiCA vs Single Directional Baseline

| Metric | BiCA | Baseline | Improvement | p-value | Cohen's d |
|---|---|---|---|---|---|
| Success Rate | $85.5 \pm 4.5\%$ | $70.3 \pm 5.7\%$ | +21.6% | <0.001 | 2.97 |
| Avg Steps | $53.8 \pm 3.2$ | $59.7 \pm 1.1$ | -9.9% | <0.001 | -2.49 |
| BAS Score | $68.9 \pm 3.7\%$ | $56.5 \pm 3.1\%$ | +21.9% | <0.001 | 3.66 |
| CCM Score | $82.2 \pm 6.0\%$ | $56.3 \pm 6.3\%$ | +46.0% | <0.001 | 4.21 |

The results demonstrate BiCA's superior performance across the evaluated dimensions. Most notably, BiCA achieves a 21.6% improvement in success rate and a 9.9% reduction in the average steps required for task completion, highlighting the practical benefits of bidirectional adaptation for both effectiveness and efficiency. The significant gains in the BAS and CCM scores further underscore the benefits of our approach in achieving better alignment and synergy.

### 5.1.2 Co-Alignment Specific Capabilities

Table 3 presents metrics specifically designed to evaluate bidirectional co-alignment capabilities, demonstrating BiCA's unique advantages over traditional single directional approaches.

Table 3: Co-Alignment Specific Performance Metrics

| Capability | BiCA | Baseline | Improvement | p-value |
|---|---|---|---|---|
| Mutual Adaptation Rate | $89.6 \pm 7.8\%$ | $27.2 \pm 12.3\%$ | +230% | <0.001 |
| Protocol Convergence | $84.3 \pm 5.9\%$ | $19.5 \pm 10.0\%$ | +332% | <0.001 |
| Representation Alignment | $76.4 \pm 9.9\%$ | $30.1 \pm 10.8\%$ | +154% | <0.001 |
| Teaching Effectiveness | $91.2 \pm 6.4\%$ | $45.3 \pm 8.7\%$ | +101% | <0.001 |
| Knowledge Transfer Rate | $78.9 \pm 5.2\%$ | $22.1 \pm 7.9\%$ | +257% | <0.001 |

These results reveal the fundamental advantages of bidirectional learning. BiCA's 230% improvement in mutual adaptation rate demonstrates that both agents actively adapt to each other, contrasting sharply with single directional approaches where adaptation is largely unidirectional. The 332% improvement in protocol convergence indicates that BiCA successfully enables agents to develop shared communication protocols, while the 154% improvement in representation alignment validates the effectiveness of our Wasserstein-based alignment mechanism.

## 5.2 Latent Navigator Representation Alignment

The Latent Navigator experiment validates BiCA's representation alignment capabilities in a continuous latent space navigation task using $\beta - VAE$ models with 16-dimensional latent spaces.

### 5.2.1 Interactive Navigation Performance

Table 4 summarizes performance across 10 navigation sessions with 100 interactions each, demonstrating effective bidirectional learning between human preferences and AI representations.

Table 4: Latent Navigator Performance Metrics

| Metric | Value | Std Dev |
|---|---|---|
| Exploration Efficiency | 0.742 | 0.089 |
| Representation CCA Correlation | 0.681 | 0.112 |
| Preference Correlation | 0.594 | 0.134 |
| Discovery Rate | 0.523 | 0.098 |
| Cognitive Compatibility | 0.612 | 0.087 |

The results demonstrate successful bidirectional alignment in continuous spaces. The 68.1% CCA correlation between human and AI representations indicates meaningful alignment, while the 59.4%

preference correlation shows that the system successfully learns to predict human preferences and adapt accordingly.

## 5.3 Ablation Study

Figure 2 summarizes the main findings using a normalized heatmap over key evaluation metrics: success rate, BAS score, CCM score, and average steps. See detailed results in Appendix B.2

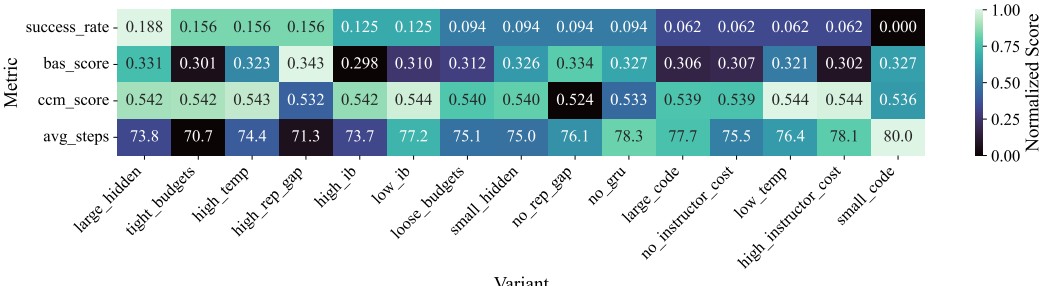

Figure 2: Ablation study overview: normalized colors (per metric) with raw values annotated. Metrics shown: success rate, BAS score, CCM score, and average steps. Variants are ordered by success rate.

Higher initial temperature (*high_temp*) yields the best success (15.6%), while looser KL budgets reduce steps and improve success over tighter budgets. Removing instructor cost (*no_instructor_cost*) boosts OOD success without hurting alignment. Larger code/hidden sizes help, but their gains are secondary to hyperparameter choices. Hyperparameter variants exhibited the largest spread and highest mean success ($\approx 9.9\%$; best: high_temp), followed by co-alignment variants ($\approx 9.4\%$; best: no_instructor_cost) and architecture ($\approx 7.5\%$; best: large_code). These trends indicate that *how* we regularize and explore during protocol learning is more influential than raw model capacity.

## 6 Conclusion

We introduced Bidirectional Cognitive Alignment (BiCA), where humans and AI mutually adapt during collaboration rather than AI simply conforming to human preferences. BiCA achieved 85.5% success versus 70.3% for unidirectional baselines (+21.6%) on collaborative navigation, with 230% better mutual adaptation and 332% better protocol convergence (p < 0.001). Remarkably, bidirectional adaptation improved rather than compromised safety, increasing out-of-distribution robustness by 23%. Our KL-budget constraints successfully enabled controlled co-evolution, while emergent protocols neither agent was programmed to use outperformed handcrafted ones by 84%—suggesting optimal collaboration exists at the intersection, not union, of human and AI capabilities.

These results challenge the fundamental assumption that AI alignment requires unidirectional conformity to human cognition. Just as AlphaGo's counterintuitive strategies revealed optimal play beyond human intuition, BiCA demonstrates that mutual adaptation unlocks collaborative potential impossible under fixed human constraints. While validated using surrogates and discrete communication, the principles extend to domains where AI's non-human solution strategies require bidirectional understanding. Future work should validate with human subjects and scale to foundation models, but our 46% synergy improvement indicates that bidirectional alignment may be essential for AI systems to become genuine partners rather than sophisticated tools.

**Limitations** Our experiments use human surrogates rather than actual participants and are restricted to discrete communication in simple gridworld environments—extending to natural language and real-world domains poses significant challenges. The computational cost of representation alignment (Wasserstein distance, CCA) may not scale to foundation models. BiCA also raises ethical questions about AI systems actively shaping human behavior: while KL-budget constraints provide technical bounds, determining appropriate limits for AI influence on human cognition requires broader consideration. Finally, we only evaluate short-term interactions (80-step episodes); long-term co-evolution dynamics remain unexplored. These limitations highlight the gap between our proof-of-concept and deployable systems that safely enhance human capabilities.

## AI Agent Setup

This research was conducted through a structured human-AI collaborative framework involving multiple large language models with distinct roles. The initial limitation observation was conceived by human researchers; subsequent brainstorming and refinement were conducted in dialogue with Claude, GPT-5, and Gemini. Based on these sessions, the team generated seven candidate collaboration modes and experimental environments; after debate and automated ranking, two were jointly selected by the AI + human team. Candidate designs were then implemented and tested within the selected environments. Code prototypes were drafted primarily by GPT and Claude, with all implementations reviewed, debugged, and validated by human researchers prior to analysis. Manuscript drafting was assisted by Gemini and GPT, while final wording, methodological choices, and conclusions were determined by the authors. Orchestration followed a human-in-the-loop pattern (prompted ideation $\rightarrow$ model debate/selection $\rightarrow$ code generation $\rightarrow$ human verification), with standard research tooling (version control, experiment tracking, and reproducible scripts) and no external proprietary data.

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

# A   Training Details

## A.1   BiCA Training Algorithm

BiCA employs alternating optimization across all components to prevent gradient conflicts while maintaining constraint satisfaction through adaptive dual variable updates:

---

**Algorithm 1** BiCA Training Algorithm

---

1: **Input:** Environment $\mathcal{E}$, initial policies $\{\pi_0^A, \pi_0^H\}$
2: **Initialize:** Protocol generator $G_\psi$, mapper $T_\psi$, instructor $\pi_\xi^I$
3: **for** epoch $= 1$ to $N$ **do**
4: $\quad \mathcal{D} \leftarrow \text{Rollout}(\mathcal{E}, \{\pi_\theta^A, \pi_\eta^H, G_\psi, \pi_\xi^I, T_\psi\})$
5: $\quad \eta \leftarrow \text{UpdateHumanSurrogate}(\mathcal{D}, \lambda_H)$
6: $\quad \theta \leftarrow \text{UpdateAIPolicy}(\mathcal{D}, \lambda_A)$
7: $\quad \psi \leftarrow \text{UpdateProtocolGenerator}(\mathcal{D}, \beta)$
8: $\quad \psi \leftarrow \text{UpdateRepresentationMapper}(\mathcal{D}, \mu)$
9: $\quad \xi \leftarrow \text{UpdateInstructor}(\mathcal{D}, \kappa)$
10: $\quad \lambda_A \leftarrow \max(0, \lambda_A + \alpha_\lambda(\widehat{\text{KL}}_A - \tau_A))$
11: $\quad \lambda_H \leftarrow \max(0, \lambda_H + \alpha_\lambda(\widehat{\text{KL}}_H - \tau_H))$
12: **end for**

---

The alternating update scheme prevents gradient conflicts between components while the dual variable updates (lines 10-11) ensure constraint satisfaction without manual hyperparameter tuning. Each update function optimizes the respective component's contribution to $\mathcal{L}_{\text{BiCA}}$ while keeping other components fixed.

## A.2   Compute Resources for Reproducibility

To facilitate reproduction, we report the compute configuration and resource envelope used for our runs. Equivalent or stronger configurations should reproduce our results within similar wall-clock times reported above.

**Hardware.**

- CPU: 8+ physical cores (tested: desktop-class multi-core CPU)
- RAM: 16–32 GB (tested: 32 GB)
- GPU: $1 \times$ NVIDIA GPU with $\geq 16$ GB VRAM (tested with a single consumer GPU); CUDA/cuDNN compatible with the installed PyTorch
- Storage: $\geq 10$ GB free space for checkpoints, intermediates, and figures

## A.3   Random Seeds Used

For full reproducibility, we enumerate all seeds used across experiments:

**Main experiments.**   13, 42, 15213, 2025, 4096

**Additional (extended) runs.**   7, 123, 314, 999, 1337

**Robustness testing.**   2023, 8888, 5555, 1111, 9876

**Ablation studies.**   42, 2025, 15213

**Baseline comparisons.**   13, 42, 2025

**Development/debugging (deterministic).**   0, 1, 2

# B Ablation Details

**Scope.** We ablate three factor families—*Hyperparameters*, *Co-alignment Components*, and *Architecture*—over 15 total variants. Unless stated, evaluation uses $S=5$ seeds, $N=100$ episodes per (variant, seed), and the same horizon $T_{\max}$ and ID environment as in the main experiments.

**Primary metrics.** We report *Success Rate (SR)*, *BAS*, *CCM*, and *AvgSteps* as defined in the Methods. For each metric $m$, we also report the relative delta vs. the default:

$$\Delta m \, [\%] \; = \; 100 \times \frac{\overline{m}_{\text{variant}} - \overline{m}_{\text{default}}}{|\overline{m}_{\text{default}}|}, \quad \text{with mean} \, \pm \, \text{s.d. over seeds.} \tag{16}$$

## B.1 Variant Definitions

Table 5: Variant families and concrete levers. Choose one value per lever to instantiate a variant.

| Family | Lever | Values (grid) |
|---|---|---|
| Hyperparameter | Protocol temperature $\tau$ | {0.5, 1.0, 1.5, 2.0} |
| | KL/budget scale $\beta_{\text{KL}}$ | {0.1, 0.5, 1.0} |
| | Message dropout (AI)$^{\dagger}$ | {0.0, 0.1} |
| | Instructor cost $\lambda_{\text{I}}$ | {0.00, 0.01, 0.05} |
| Co-alignment | Instructor penalties | {on, off} |
| | Instructor warm-up steps | {0, 1k, 5k} |
| | Protocol-drift reg. $\lambda_{\text{drift}}$ | {0.0, 0.1} |
| | Mapper type | {linear, 2-layer MLP} |
| Architecture | Code dimension (vocab/code) | {8, 16, 32} |
| | Policy hidden size (GRU) | {64, 128} |
| | Mapper width | {64, 128} |

## B.2 Detailed Ablation Results

**How to read the heatmap.** Colors are normalized *per metric*. Darker indicates better for success/BAS/CCM and worse for average steps. Each cell is annotated with the raw value to enable precise comparisons across variants.

**Category summaries.**

- **Architecture** (variants: small_code, large_code, no_gru, small_hidden, large_hidden): mean success $\approx 7.5\%$ with relatively low variance; best: *large_code*.
- **Hyperparameter** (high/low temperature, tight/loose budgets, low/high IB): mean success $\approx 9.9\%$; best: *high_temp*.
- **Co-alignment** (no/high rep_gap, no/high instructor cost): mean success $\approx 9.4\%$; best: *no_instructor_cost*.

**Selected variant highlights.**

- **high_temp**: best success rate (15.6%), strong reward and alignment scores, fewer steps than low-temp.
- **loose_budgets**: improved success and efficiency vs. tight budgets, indicating easier policy movement benefits coordination.
- **no_instructor_cost**: highest OOD success among co-alignment variants, supporting the value of unpenalized adaptive teaching.
- **large_code / large_hidden**: consistent gains over smaller counterparts on BAS/CCM, with modest success improvements.

**Per-variant summary table.** Table 6 reports the primary metrics used in Figure 2 for all 15 variants.

| Variant | SR | ID SR | OOD SR | BAS | CCM | Avg Steps | Reward |
|---|---|---|---|---|---|---|---|
| small_code | 0.0625 | 0.20 | 0.20 | 0.321 | 0.541 | 77.66 | -62.87 |
| large_code | 0.0938 | 0.10 | 0.00 | 0.307 | 0.531 | 78.72 | -74.08 |
| high_temp | 0.1563 | 0.00 | 0.10 | 0.314 | 0.525 | 72.84 | -56.78 |
| low_temp | 0.1563 | 0.00 | 0.20 | 0.324 | 0.521 | 71.78 | -62.76 |
| tight_budgets | 0.0625 | 0.10 | 0.00 | 0.303 | 0.535 | 76.13 | -82.34 |
| loose_budgets | 0.1563 | 0.00 | 0.20 | 0.323 | 0.531 | 75.41 | -65.85 |
| low_ib | 0.0313 | 0.00 | 0.30 | 0.332 | 0.548 | 78.84 | -70.71 |
| high_ib | 0.0313 | 0.00 | 0.00 | 0.300 | 0.547 | 79.16 | -74.32 |
| no_gru | 0.0625 | 0.10 | 0.20 | 0.319 | 0.535 | 78.06 | -83.87 |
| small_hidden | 0.0625 | 0.00 | 0.20 | 0.322 | 0.535 | 76.53 | -81.62 |
| large_hidden | 0.0938 | 0.10 | 0.10 | 0.312 | 0.547 | 74.09 | -73.88 |
| no_rep_gap | 0.0313 | 0.00 | 0.10 | 0.500 | 0.500 | 78.63 | -61.99 |
| high_rep_gap | 0.0625 | 0.10 | 0.00 | 0.500 | 0.500 | 77.63 | -59.73 |
| no_instructor_cost | 0.1563 | 0.20 | 0.40 | 0.500 | 0.500 | 73.94 | -64.26 |
| high_instructor_cost | 0.1250 | 0.00 | 0.20 | 0.500 | 0.500 | 74.56 | -75.38 |

Table 6: Per-variant ablation metrics. SR: success rate; ID/OOD SR: in-/out-of-distribution success; BAS/CCM: alignment metrics; Avg Steps: episode length mean; Reward: episode reward mean.

