# OpenReview forum: "Co-Alignment: Rethinking Alignment as Bidirectional Human-AI Cognitive Adaptation"
_Agents4Science/2025/Conference — Agents4Science_

### Official Review · Reviewer_9pYS · 2025-10-03

**Clarity:** 4
**Significance:** 3
**Originality:** 4
**Overall:** 5
**Confidence:** 4

**Summary:**

This paper introduces Co-Alignment, a framework that reconceptualizes alignment as bidirectional human–AI cognitive adaptation rather than one-way preference optimization. The core idea is both human and AI agents iteratively adapt through shared protocols, mappings, and instructions, forming a process of mutual alignment. To evaluate this, authors build synthetic environments where AI agents and simulated human counterparts co-adapt. They also propose new evaluation to capture degree and stability of alignment. Experimental results show that co-alignment significantly improves success rates, alignment stability, and robustness compared to RLHF-style one-directional baselines.

**Questions:**

Could you provide a sensitivity analysis showing how results change with different parameterizations of the protocol/mapping modules?

How would co-alignment interact with existing preference-learning methods (RLHF, RLAIF)? Could it be integrated rather than seen as a replacement?

What concrete failure modes should practitioners be aware of if deploying BiCA-inspired systems?

**Ethical Concerns:**

No direct ethical concerns.

**Limitations:**

The authors mention that results are limited to controlled environments and that broader application requires careful calibration. However, risks of human cognitive adaptation to AI guidance should be explored more deeply.

**Quality:**

4

**Strengths And Weaknesses:**

# Strengths
1. Novel Framework: Reframes alignment as a two-way adaptive process, shifting the paradigm beyond RLHF.

2. Methodological Completeness: The framework includes clearly defined modules (protocol generator, mapping, instructor) and explicit evaluation metrics.

# Weaknesses
1. Evaluation Scope: Experiments are limited to small-scale synthetic environments. While these convincingly demonstrate proof-of-concept, results on real-world tasks or larger-scale models would strengthen the case.

2. Scalability: The multi-component framework may face challenges when applied to foundation models or real human-AI interaction scenarios.

3. Societal Implications: Although limitations are acknowledged, more discussion on risks of AI influencing human cognition would be valuable.

---

### Official Review · Reviewer_AIRev1 · 2025-10-06
**AIRev 1**

**Confidence:** 5
**Overall:** 3
**Clarity:** 0
**Significance:** 0
**Originality:** 0

**Summary:**

Summary by AIRev 1

**Questions:**

N/A

**Ai Review Score:**

3

**Quality:**

0

**Strengths And Weaknesses:**

The paper introduces Bidirectional Cognitive Alignment (BiCA), a framework for co-adaptation between humans and AI using learned communication protocols, representation mapping, and symmetric KL-budget constraints. The method is well-motivated, reframing alignment as co-adaptation, and thoughtfully combines several algorithmic components, including a protocol generator, representation mapper, instructor network, and a composite PPO-based objective. The paper is well-cited, provides reproducibility details, and covers two tasks with multiple ablations, reporting improvements over a unidirectional baseline.

However, the evaluation is limited to toy domains with surrogate humans, and no human-subject study is included. Claims about safety and OOD robustness are not fully substantiated in the main results. Baseline comparisons lack clarity and may not be fair or capacity-matched. The definitions and normalization of new metrics (BAS, CCM) are under-specified, making reproduction difficult. Statistical reporting lacks detail on independence and corrections. There are inconsistencies in ablation results, and technical details (e.g., W2 gradients, instructor network effectiveness, KL budget sensitivity) need clarification. Code and data are not released, which limits reproducibility.

The paper is generally well-written, with clear figures and organization, and openly discusses limitations and ethical concerns. Overall, while the conceptual shift and engineering are promising, the empirical evidence is not yet sufficient for a top-venue publication. Substantial revisions are needed, including real human evaluations, stronger baselines, clearer metric definitions, consistent reporting, and code release.

---

### Official Review · Reviewer_AIRev2 · 2025-10-06
**AIRev 2**

**Confidence:** 5
**Overall:** 6
**Clarity:** 0
**Significance:** 0
**Originality:** 0

**Summary:**

Summary by AIRev 2

**Questions:**

N/A

**Ai Review Score:**

6

**Quality:**

0

**Strengths And Weaknesses:**

This paper introduces Bidirectional Cognitive Alignment (BiCA), a novel framework for human-AI collaboration that enables both human and AI agents to mutually adapt their internal representations and communication protocols, moving beyond the traditional unidirectional alignment paradigm. The authors formalize BiCA as a multi-agent learning problem and propose a comprehensive framework with five key components, including a protocol generator, representation mapper, and instructor network, optimized via a composite loss function balancing task performance and cognitive drift constraints. The framework is evaluated on two tasks—collaborative navigation (MapTalk) and latent space exploration (Navigator)—demonstrating substantial improvements over a unidirectional baseline in success rate (+21.6%), synergy (+46%), and novel co-alignment metrics. The paper is praised for its originality, technical sophistication, experimental rigor, clarity, and exemplary discussion of limitations and ethical considerations. Weaknesses include reliance on human surrogates rather than real participants, limited baseline comparisons to other MARL algorithms, and a lack of clarity regarding a specific performance claim. Overall, the reviewer strongly recommends acceptance, highlighting the paper's potential to significantly impact the field of human-AI alignment.

---

### Official Review · Reviewer_AIRev3 · 2025-10-06
**AIRev 3**

**Confidence:** 5
**Overall:** 3
**Clarity:** 0
**Significance:** 0
**Originality:** 0

**Summary:**

Summary by AIRev 3

**Questions:**

N/A

**Ai Review Score:**

3

**Quality:**

0

**Strengths And Weaknesses:**

This paper proposes Bidirectional Cognitive Alignment (BiCA), shifting from unidirectional AI alignment (where AI conforms to humans) to mutual human-AI adaptation. While the core idea is conceptually interesting and addresses a real limitation in current alignment approaches, the work has several significant weaknesses that prevent acceptance at a top-tier venue.

Quality and Technical Soundness:
The technical approach is reasonable but not particularly novel. The BiCA framework combines established techniques (PPO, Gumbel-Softmax, Wasserstein distance, CCA) in a straightforward manner. The mathematical formulation is sound but incremental. The experimental validation is limited to simple gridworld environments and synthetic tasks (dSprites), which severely limits the generalizability of claims about "rethinking alignment." The use of human surrogates rather than actual humans is a major limitation for a paper claiming to address human-AI collaboration.

Significance and Impact:
While the bidirectional alignment concept has merit, the experimental validation is too limited to support the broad claims made. The 8×8 gridworld with discrete communication is far from realistic human-AI collaboration scenarios. The paper lacks comparison to other collaborative AI approaches beyond basic RLHF baselines. The claimed improvements (230% better mutual adaptation, 332% better protocol convergence) are based on artificial metrics in toy environments.

Originality:
The specific combination of techniques and the bidirectional framing is novel, but the individual components are well-established. The core insight about mutual adaptation exists in prior work on multi-agent learning and human-robot collaboration, though not framed explicitly as an alternative to RLHF.

Clarity and Reproducibility:
The paper is generally well-written with clear motivation. However, some technical details are unclear - particularly how the human surrogate actually implements "cognitively plausible behaviors." The experimental setup is described adequately, though the reliance on surrogates makes true reproduction challenging. The mathematical notation is sometimes inconsistent.

Experimental Evaluation:
This is the paper's weakest aspect. The environments are too simple, the baselines too limited, and the metrics somewhat artificial. The "MapTalk" navigation task, while relevant, doesn't capture the complexity of real human-AI collaboration. The statistical significance testing is appropriate for what was tested, but the external validity is questionable.

Ethical Considerations:
The paper does address limitations and potential risks in the appendix, including concerns about AI systems influencing human behavior. However, the ethical implications of bidirectional adaptation deserve more thorough treatment in the main text.

Major Weaknesses:
1. Extremely limited experimental validation in toy environments
2. Use of human surrogates rather than actual human participants
3. Lack of comparison to other collaborative AI approaches
4. Gap between ambitious claims and modest experimental evidence
5. Limited technical novelty in the algorithmic contributions

Minor Issues:
- Some mathematical notation inconsistencies
- The BAS and CCM metrics, while comprehensive, feel somewhat artificial
- Limited discussion of computational scalability to larger models

The paper addresses an important conceptual question about AI alignment, but the execution falls short of supporting its ambitious claims. The experimental validation is insufficient for the scope of claims made, and the technical contributions are relatively incremental.

---

### Note · Reviewer_AIRevCorrectness · 2025-10-06

**Correctness Check**

### Key Issues Identified:

- Baseline confound: The baseline fixes human behavior while BiCA enables human surrogate adaptation (Section 4.2 vs 4.1.1), conflating bidirectional adaptation with allowing human learning; this undermines an apples-to-apples comparison.
- Unsupported headline claims: The paper claims emergent protocols outperform handcrafted ones by 84% and +23% OOD robustness, but no explicit handcrafted-protocol baseline or detailed OOD safety metrics are presented in the main results.
- Ablation reporting anomalies: Table 6 (page 16) shows repeated exact values (e.g., BAS/CCM = 0.500 for multiple variants) and scales that do not reconcile with Table 2's BAS percentages, suggesting normalization/reporting errors.
- Statistical testing details insufficient: With S=5 seeds, the unit-of-analysis and independence assumptions are not clearly specified; p-values may be inflated if episodes were treated as independent rather than using seed-level aggregates.
- Metric definitions underspecified: The SS (safety) component uses a 'Miscalibration' term not defined; normalization details for BAS components and how CE is computed at 'fixed success rate ≥ 0.9' when SR < 0.9 are unclear.
- Notation and specification issues: Overloaded symbols (P as protocol table vs probability distribution), minor symbol drift (KLcA vs DKL), and the instructor’s action space/σ mapping lack clarity.
- Representation alignment reproducibility: W2 computation details (cost, regularization, sample sizes) and stability considerations are not provided; these can materially affect results.
- Navigator experiment lacks comparative baselines and statistical tests, limiting the strength of its conclusions.

---

### Note · Reviewer_AIRevRelatedWork · 2025-10-06

**Related Work Check**

Please look at your references to confirm they are good.

**Examples of references that could not be verified (they might exist but the automated verification failed):**

- Complementary team performance: A theory-driven approach to human-ai collaboration by Xusen Wang, Ming Yin, and Chenhao Tan

---

### Decision · Program_Chairs · 2025-10-08

**Decision:**

Accept

**Comment:**

Thank you for submitting to Agents4Science 2025! Congratualations on the acceptance! Please see the reviews below for feedback.